

# A lacustrine surface-sediment pollen dataset covering the Tibetan Plateau and its potential in past vegetation and climate reconstructions

**Fang Tian[1], Weiyu Cao[1], Xiaohan Liu[1], Zixin Liu[1], Xianyong Cao[2]**

[1] *College of Resource Environment and Tourism, Capital Normal University, Beijing 100048, China*

[2] *Group of Alpine Paleoecology and Human Adaptation (ALPHA), State Key Laboratory of Tibetan Plateau Earth System, Environment and Resources (TPESER), Institute of Tibetan Plateau Research, Chinese Academy of Sciences, Beijing 100101, China*

Correspondence: Fang Tian (tianfang@cnu.edu.cn)

**Abstract.** A dataset of pollen extracted from the surface-sediments of lakes with an even spatial distribution is essential for pollen-based reconstructions of past vegetation and climate. We collected 90 lake surface-sediment samples from the Tibetan Plateau (TP) covering major vegetation types. A comprehensive modern pollen dataset is established by integrating our newly obtained modern pollen dataset with previous modern lacustrine pollen datasets, covering the full range of climatic gradients across the TP with mean annual precipitation ($P_{ann}$) from 97 to 788 mm, mean annual temperature ($T_{ann}$) -9.09 to 6.93 °C, mean temperature of the coldest month ($Mt_{co}$) -23.48 to -2.65°C, and mean temperature of the warmest month ($Mt_{wa}$) 1.77 to 19.26°C. Numerical analyses revealed that $P_{ann}$ is the primary climatic determinant for pollen distribution, while net primary production (NPP) is a valuable variable reflecting vegetation conditions. To detect the quantitative relationship between pollen and $P_{ann}$/NPP, both weighted-averaging partial least squares (WA-PLS) and random forest algorithm (RF) were employed. The performance of both models suggests that this modern pollen dataset has good predictive power in estimating past NPP and $P_{ann}$, but RF has a slight advantage with this dataset. This comprehensive modern pollen dataset is considered reliable when reconstructing vegetation and climate from pollen spectra from the central TP, but caution is needed if it is applied to pollen spectra from the marginal regions of the TP and those covering the Last Glacial period, due to poor analogue quality in those cases. The dataset, including site locations, pollen percentages, NPP, and climate data for 90 lakes, is available at the National Tibetan Plateau Data Center (TPDC; Tian, 2025; https://doi.org/10.11888/Paleoenv.tpdc.302470).



# 1 Introduction

A modern pollen dataset is the foundation for the quantitive reconstruction of past vegetation and climate based on fossil pollen spectra. Surface-soil samples for pollen analysis can be easily obtained, but their pollen assemblages are easily affected by local vegetation components, which cause more noise in the modern relationships of pollen–climate and pollen–vegetation (Cao et al., 2014). Sediment from lakes, in contrast, provide more regional pollen signals owing to broader pollen source areas, more stable sedimentation rates, and better preservation, making them more suitable for regional vegetation and climate changes (Tian et al., 2020; Cao et al., 2021). Due to the sparse distribution of lakes, high sampling costs, and limited accessibility—especially in remote regions—modern pollen datasets from lake surface sediments remain limited and spatially biased, particularly in China (Herzschuh et al., 2010; Ma et al., 2017; Cao et al., 2021).

Situated at high elevations and subject to complex climate systems, the Tibetan Plateau (TP) is highly sensitive to global climate change and human activities and exhibits strong regional ecological and climatic peculiarities (Chen et al., 2015, 2020; Pepin et al. 2019). These features make the TP a research hotspot for past climate and vegetation reconstructions. Fortunately, the widespread distribution of lakes across the plateau offers an opportunity to expand and refine pollen-based calibration datasets using lake surface sediments., but the distribution of available pollen datasets of lake surface-sediment remains uneven and incomplete  due to logistical constraints (Cao et al., 2021; Qin, 2021; Ma et al., 2024). Hence, it is essential to improve the coverage and comprehensiveness of the modern calibration-set from lake surface-sediments on the TP.

Previous pollen–climate relationships are often the focus of calibration-set studies, while the pollen–vegetation relationship is also crucial on the TP, where vegetation type is generally employed as the target variable, especially when reconstructing ecological conditions (e.g. Qin, 2021, Qin et al., 2022). However, the available modern pollen datasets reveal that pollen assemblages from different vegetation types on the TP generally present only minor differences in pollen components and their abundance. For instance, the dominant pollen taxa are generally herbaceous taxa including Cyperaceae, *Artemisia*, Amaranthaceae (=Chenopodiaceae), and Poaceae (e.g. Herzschuh et al., 2010; Ma et al., 2017; Cao et al., 2014, 2021; Li et al., 2020; Qin, 2021), making it difficult to distinguish vegetation conditions based on pollen assemblages directly. Net primary production (NPP), which quantifies the amount of atmospheric carbon fixed by plants and accumulated as biomass, plays an important role in the global carbon cycle (Fang et al., 2001; Nemani et al., 2003; Gonsamo et al., 2013; Ni, 2013; Walker et al., 2015; Ji et al., 2020). Therefore, NPP may serve as a





more sensitive alternative variable in reflecting the spatial distribution and temporal
change of vegetation conditions on the TP.

Here, we analysed 90 lake surface-sediment samples for pollen and combined them
with previously published modern pollen data extracted from lake surface-sediments
(Herzschuh et al., 2010; Li and Li, 2015; Cao et al., 2021; Ma et al., 2024; Wu et al.,
2024), then used a combination of ordination techniques, weighted averaging partial
least squares (WA-PLS), and Random Forest (RF) to: (1) establish a comprehensive
pollen dataset extracted from lake surface-sediments covering the entire TP with an
even distribution; (2) evaluate the predictive power of models using the modern pollen
dataset in reconstructing past climate and vegetation.

## 2    Study area

Climate of the TP is controlled mainly by the Asian Summer Monsoon in summer with
warm-wet conditions and by westerlies in winter with a cold-dry climate (Wang, 2006).
In addition, there is a gradient from high summer temperatures (up to 19°C) and high
precipitation (>700 mm) on the south-eastern TP, to low summer temperatures (ca. 6°C)
and low precipitation (<100 mm) on the north-western TP (Fig. 1; Sun, 1999;
Herzschuh, 2007; He et al., 2020).

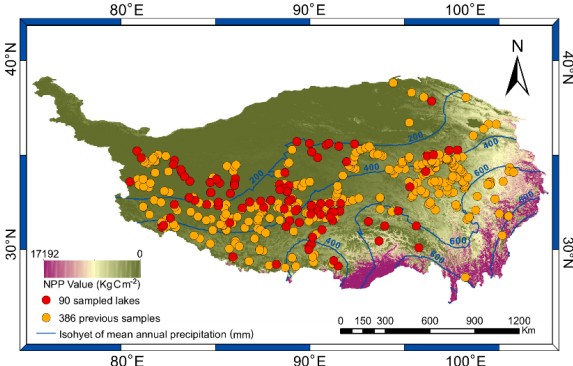

**Figure 1**. Spatial distribution of 476 modern pollen samples collected from lake surface-sediments
on the Tibetan Plateau (red filled circles: 90 sampled lakes; orange filled circles: 386 previous
samples, Herzschuh et al., 2010; Li and Li, 2015; Cao et al., 2021; Ma et al., 2024; Wu et al., 2024).

The TP exhibits distinct vegetation zonation along its south-east–north-west thermal
and moisture gradients, progressing from forest ecosystems through alpine meadows
and steppes to desert vegetation (Fig. 1; Zhang, 2007). Forests dominated by *Pinus*,
*Picea*, *Abies*, *Betula*, *Quercus*, and *Tsuga* are primarily distributed in the warm-humid
south-eastern and eastern marginal regions of the TP (Herzschuh, 2007). Alpine
meadows, as one of the most important vegetation types, mainly distributed on the
eastern and southern TP, and are characterized by *Kobresia* spp., *Carex*, Asteraceae,





*Polygonum*, *Potentilla*, Fabaceae, Caryophyllaceae, *Leontopodium*, *Arenaria*,
*Ranunculus,* and Poaceae (Wu, 1995; Herzschuh et al., 2010; Cao et al., 2021). Alpine
steppes are primarily distributed across the southern, eastern, and central TP, and is
mainly dominated by *Stipa purpurea*, *Artemisia*, *Potentilla*, Asteraceae, Amaranthaceae,
and *Carex* (Fig. 1; Zhang, 2007; Yue et al., 2011). Alpine deserts, located in the dry
north-central and westernmost central TP, are characterized by sparse vegetation, and
are predominantly occupied by drought-tolerant taxa such as *Ceratoides*
(Amaranthaceae), *Salsola*, *Haloxylon*, *Kalidium*, *Artemisia*, *Ephedra*, *Nitraria,* and
Poaceae (Fig. 1; Zhang, 2007).
**3  Materials and methods**
3.1  Sample collection and pollen processing
To ensure the even distribution of the sampled lakes, we collected lake surface-
sediment samples (the top 2 cm, *n*=90) from the centre of each lake in forest (*n*=5),
meadow (*n*=22), steppe (*n*=53), and desert (*n*=10) vegetation types on the TP between
2021 and 2023 (Fig. 1, Table 1). The elevation range of these lakes varies from 3923 to
5433 m a.s.l. with a median of 4652 m a.s.l. (Fig. 1).
**Table 1.** Locations of the sampling sites of our field work on the Tibetan Plateau.

| No. | Lake | Latitude (°N) | Longitude (°E) | Elevation (m a.s.l.) | Vegetation type |
|---|---|---|---|---|---|
| 1 | Cuomujiri | 94.4304 | 29.8118 | 4235 | Forest |
| 2 | Ranwu Lake | 96.8252 | 29.3962 | 5263 | Forest |
| 3 | Sanse Lake | 94.7670 | 30.7239 | 4042 | Forest |
| 4 | Ren Co | 96.6748 | 30.7156 | 4452 | Forest |
| 5 | Potal Lake | 95.5743 | 31.6223 | 4656 | Forest |
| 6 | Ruba Lake | 90.1725 | 29.4644 | 3923 | Meadow |
| 7 | Namucoluo | 90.3347 | 29.6070 | 4690 | Meadow |
| 8 | Cuoriwang | 90.4064 | 30.0345 | 4400 | Meadow |
| 9 | Niangde Co | 90.1834 | 29.2810 | 4365 | Meadow |
| 10 | Cona Lake | 91.4305 | 32.0779 | 4602 | Meadow |
| 11 | Tangbin Lake | 90.9672 | 30.4795 | 5025 | Meadow |
| 12 | Cuoe | 91.5350 | 31.5088 | 4511 | Meadow |
| 13 | Changma Lake | 92.1069 | 32.0639 | 4932 | Meadow |
| 14 | Cuomuri | 92.0596 | 31.6201 | 4547 | Meadow |
| 15 | Gemu Co | 91.6990 | 31.5550 | 4524 | Meadow |
| 16 | Xiongmu Co | 91.6303 | 31.0399 | 4662 | Meadow |
| 17 | Nairi Pingco | 91.4788 | 31.2730 | 4513 | Meadow |
| 18 | Cuomuzhelin | 88.2168 | 28.3933 | 4395 | Meadow |
| 19 | Nariyong Co | 91.9377 | 28.3071 | 4731 | Meadow |
| 20 | Peiku Co | 85.5869 | 28.8507 | 4561 | Meadow |
| 21 | Zhegu Co | 91.6770 | 28.6316 | 4601 | Meadow |
| 22 | Nianjie Co | 96.2905 | 33.0773 | 4441 | Meadow |



| 23 | Samu Co | 93.7813 | 30.9753 | 4748 | Meadow |
|---|---|---|---|---|---|
| 24 | Haling Lake | 97.5967 | 38.2507 | 4071 | Meadow |
| 25 | Zhaling Lake | 97.3420 | 34.9447 | 4280 | Meadow |
| 26 | Koucha Lake | 97.2311 | 34.0081 | 4518 | Meadow |
| 27 | Eling Lake | 97.7130 | 35.0217 | 4257 | Meadow |
| 28 | Gelu Co | 92.4546 | 34.5942 | 4639 | Steppe |
| 29 | UlanUl Lake | 90.7108 | 34.8528 | 4857 | Steppe |
| 30 | Xijir Ulan Lake | 90.3528 | 35.1875 | 4769 | Steppe |
| 31 | Lexiewudan Lake | 90.2053 | 35.7071 | 4862 | Steppe |
| 32 | Xiangyang Lake | 89.4616 | 35.8194 | 4843 | Steppe |
| 33 | Kekexili Lake | 91.2205 | 35.6115 | 4875 | Steppe |
| 34 | Kekao Lake | 91.3874 | 35.6973 | 4881 | Steppe |
| 35 | Zhuonai Lake | 91.9833 | 35.5325 | 4734 | Steppe |
| 36 | Kusai Lake | 92.9412 | 35.6753 | 4471 | Steppe |
| 37 | Zigêtang Co | 90.8973 | 32.0674 | 4538 | Steppe |
| 38 | Daru Co | 90.7324 | 31.6562 | 4675 | Steppe |
| 39 | Bange Lake | 89.4734 | 31.7282 | 4519 | Steppe |
| 40 | Lingge Co | 88.7220 | 33.9370 | 5061 | Steppe |
| 41 | Qiagang Co | 88.3966 | 33.2313 | 4719 | Steppe |
| 42 | Caiduochaka Lake | 88.9793 | 33.1576 | 4833 | Steppe |
| 43 | Eya Co | 88.6713 | 33.0013 | 4824 | Steppe |
| 44 | Ri Co | 89.6068 | 30.9302 | 4648 | Steppe |
| 45 | Mujiu Co | 89.0144 | 31.0337 | 4664 | Steppe |
| 46 | Suo Co | 90.9056 | 31.3978 | 4556 | Steppe |
| 47 | Mading Co | 90.2995 | 31.4147 | 4680 | Steppe |
| 48 | Maiding Co | 90.3202 | 31.8413 | 4773 | Steppe |
| 49 | Changma Co | 87.8756 | 32.2605 | 4725 | Steppe |
| 50 | Cuolongjiao | 88.8539 | 32.7857 | 4873 | Steppe |
| 51 | Duomaxiang Lake | 89.1268 | 32.3249 | 4704 | Steppe |
| 52 | Gewa Co | 88.7968 | 30.6725 | 4745 | Steppe |
| 53 | Wojiong Co | 89.3646 | 31.6276 | 4598 | Steppe |
| 54 | Gaa Co | 88.9583 | 32.2130 | 4602 | Steppe |
| 55 | Chelachapuka | 86.1548 | 31.8024 | 4773 | Steppe |
| 56 | Yong Co | 84.7044 | 31.9383 | 4712 | Steppe |
| 57 | Rena Co | 84.2559 | 32.7281 | 4579 | Steppe |
| 58 | Chabo Co | 84.2108 | 33.3512 | 4500 | Steppe |
| 59 | Jibuchaga Co | 83.9975 | 32.0205 | 4467 | Steppe |
| 60 | Cuoguo Co | 83.2921 | 32.2503 | 4669 | Steppe |
| 61 | Bieruoze Co | 82.9417 | 32.4308 | 4392 | Steppe |
| 62 | Shekazhi | 82.0466 | 32.0115 | 4591 | Steppe |
| 63 | Dagze Co | 87.4456 | 31.8332 | 4465 | Steppe |
| 64 | Xiabie Co | 87.2680 | 32.2179 | 4592 | Steppe |
| 65 | Jiaruo Co | 86.6001 | 32.1730 | 4445 | Steppe |
| 66 | Xuguo Co | 90.3251 | 31.9542 | 4598 | Steppe |





| 67 | Beilei Co | 88.4296 | 32.9120 | 4797 | Steppe |
|---|---|---|---|---|---|
| 68 | Unknown | 81.7962 | 31.1937 | 5433 | Steppe |
| 69 | Nading Co | 85.4359 | 32.6776 | 4845 | Steppe |
| 70 | Bala Co | 82.9849 | 33.4281 | 4757 | Steppe |
| 71 | Dong Co | 84.7120 | 32.1440 | 4388 | Steppe |
| 72 | Xiaogemu Co | 85.7384 | 33.5778 | 4711 | Steppe |
| 73 | Ningri Co | 85.6752 | 33.3333 | 5020 | Steppe |
| 74 | Guping Lake | 85.6787 | 33.1683 | 5030 | Steppe |
| 75 | Qiuruba Co | 84.7966 | 33.3073 | 4733 | Steppe |
| 76 | Caima'er Co | 84.5879 | 33.5469 | 4573 | Steppe |
| 77 | Selin Co | 88.6979 | 31.7363 | 4512 | Steppe |
| 78 | Zhari Namco | 85.4004 | 30.9068 | 4595 | Steppe |
| 79 | Kuhai Lake | 99.1636 | 35.3070 | 4117 | Steppe |
| 80 | Donggi Cona | 98.6596 | 35.2875 | 4066 | Steppe |
| 81 | Aru Co | 82.4768 | 33.9682 | 4904 | Desert |
| 82 | Aksai Chin Lake | 79.7863 | 35.2456 | 4831 | Desert |
| 83 | Kunchuke Co | 82.6590 | 33.7096 | 5042 | Desert |
| 84 | Xiawei Lake | 82.0454 | 34.6738 | 5110 | Desert |
| 85 | Luotuo Lake | 81.9849 | 34.4339 | 5082 | Desert |
| 86 | Meima Co | 82.4404 | 34.1278 | 4897 | Desert |
| 87 | Lhanag Co | 81.2820 | 30.6674 | 4577 | Desert |
| 88 | Hongshan Lake | 80.0545 | 34.8300 | 5043 | Desert |
| 89 | Manasarovar Lake | 81.3939 | 30.7465 | 4577 | Desert |
| 90 | Xiada Co | 79.3584 | 33.3916 | 4338 | Desert |

For each sample, 2–3 g of dry material was used for pollen extraction, and a tablet
with *Lycopodium* spores (10,315 grains) was added to each sample initially as tracers
(Maher, 1981). Pollen samples were processed using standard acid–alkali–acid
procedures (Fægri and Iversen, 1989), including 10% HCl, 10% KOH, 40% HF,
acetolysis treatment, and sieving in an ultrasonic bath to remove particles <7 μm. Pollen
grains were identified and counted under a Zeiss optical microscope at 400×
magnification, referring to modern pollen slides collected from the eastern and central
TP and published palynological literatures (Wang et al., 1995; Tang et al., 2016; Cao et
al., 2020). To ensure the reliability of the pollen assemblages for numerical analyses,
more than 500 terrestrial pollen grains, or over 2000 *Lycopodium* spores were counted
for each sample. The pollen diagram was constructed using Tilia software (Grimm,

1987, 1991).

3.2    Data collection and harmonization
We compiled a dataset of modern pollen assemblages from lake surface sediments
across the Tibetan Plateau, incorporating 375 lakes situated in eastern (Herzschuh et al.,
2010; Cao et al., 2021), central, and western TP (Ma et al., 2024; Wu et al., 2024),
obtained from accessible databases or from authors directly. To enhance spatial



coverage, an additional 11 surface pollen assemblages were digitized from published
diagram representing sites along the eastern edge of TP (Li and Li, 2015). The total
dataset comprises 476 pollen assemblages from lake surface-sediments on the TP (Fig.
A1).

The pollen data are standardized following the procedures outlined in Cao et al.
(2013), including harmonization of taxonomy – generally to the family or genus level
– and recalculation of pollen percentages based on total terrestrial pollen grains. Only
pollen taxa with an abundance of at least 0.5% in at least three samples and a maximum
$\geq$3% were retained for statistical analyses (*n*=35).

We employed the Chinese Meteorological Forcing Dataset (CMFD), a gridded near-
surface meteorological dataset covering the period from January 1979 to December
2018, with a temporal resolution of 3 h and a spatial resolution of 0.1°. Climate data of
each sampled lake were assigned as the values of the nearest pixel from the
meteorological dataset. For each lake, the following parameters were extracted: $P_{ann}$:
mean annual precipitation, mm; $T_{ann}$: mean annual temperature, °C; $Mt_{co}$: mean
temperature of the coldest month, °C; $Mt_{wa}$: mean temperature of the warmest
month, °C (He et al., 2020). The geographical distances between lake coordinates and
grid centroids were calculated geodetically using the the *rdist.earth* function in the
*fields* package version 16.3.1 for R (R Core Team, 2019; Nychka et al., 2025).

The NPP value, defined as Gross Primary Productivity (GPP) minus Maintenance
Respiration (MR) (Zhao and Running, 2010), was obtained from observations of the
MOD17A3HGF.006 product during 2001–2022 with a pixel resolution of 1000 m.
Across the study region, NPP values range from 0.16 to 6617.36 Kg C m$^{-2}$, $P_{ann}$ ranging
from 97 to 788 mm, and cold thermal conditions characterized by low $T_{ann}$ (-9.09 to
6.93°C) and $Mt_{co}$ (-23.48 to -2.65°C; Table 2).
3.3    Data analysis

To visualize how the modern pollen assemblages respond to climatic variables,
ordination techniques were employed. Pollen data were square-root transformed to
stabilize variances and optimize the signal-to-noise ratio (Prentice, 1980). Detrended
correspondence analysis (DCA; Hill and Gauch, 1980) showed that the gradient length
of the first axis of the pollen data was 2.36 SD (standard deviation units), indicating
that a linear response model is suitable for our pollen dataset (ter Braak and
Verdonschot, 1995). We employed redundancy analysis (RDA) to assess how major
pollen taxa and sampling sites are distributed along climate gradients. Climatic
predictors were introduced sequentially following a forward selection procedure, with
multicollinearity assessed at each step via variance inflation factors (VIF). Variables
exhibiting VIF values above the threshold of 20 were excluded to maintain model
parsimony and reduce redundancy (ter Braak and Prentice, 1988; Birks, 1995).
Additionally, the suitability of each climatic variable for quantitative reconstruction was
evaluated using the ratio of the first constrained eigenvalue to the first unconstrained
eigenvalue ($\lambda 1/\lambda 2$), where larger ratios indicate stronger predictive potential (Juggins,
2013). All ordinations were carried out using the *rda* and *decorana* functions in the
*vegan* package (Oksanen et al., 2019).
WA-PLS regression was applied to calibrate transfer functions linking modern pollen
assemblages to Pann and NPP, based on square-root transformed relative abundances
of the 35 selected taxa—consistent with those used in the ordination analyses (ter Braak
and Juggins, 1993). Model performance was evaluated using "leave-one-out" cross-
validation, and the optimal number of WA-PLS components were determined based on
a randomization *t*-test (Juggins and Birks, 2012). All the analyses were performed using
the *WA-PLS* function of the *rioja* package version 0.7–3 (Juggins, 2012) in R.
As WA-PLS is known to produce systematic prediction biases near the ends of
environmental gradients—commonly referred to as the "edge effect" (Birks, 1998; Tian
et al., 2022) —we further explored a complementary reconstruction method. Random
Forest (RF) is an ensemble learning algorithm that integrates multiple decision trees
based on a classification tree algorithm and summarizes their results for classification
or regression tasks (Breiman, 2001). The importance of the explanatory variable is
normally measured as a percentage increase in the residual sum of squares after random
shuffling of the order of the variables, thereby determining which explanatory variable
can be added to the model. RF has been applied in the geographical and ecological
fields and performs well (Li, 2013; Jin et al., 2016). In this study, we applied RF to
establish the importance of pollen and the NPP/climate variables (Table S1). The model
was systematically optimized through a stepwise reduction procedure, in which the
pollen taxa with the least important score was deleted until the RF-importance of all
remaining taxa were greater than 0 (Breiman, 2001). The RF algorithm was run based
on square-root transformed pollen percentages, using the *randomForest* function in the
*randomForest* package version 4.6–14 (Liaw, 2018) in R. The statistical significance
of the reconstructions derived from WA-PLS and RF were tested with the *randomTF*
function of the *palaeoSig* package (Telford and Birks, 2011; Telford, 2013) in R.
In quantitative climate reconstructions, the taxonomic distance between a fossil
pollen assemblage and its modern analogue is a key variable in evaluating the analogue
quality (Birks et al., 1990). Shorter distances indicate closer taxonomic similarity and
higher analogue quality, enhancing reconstruction reliability. This distance is
commonly calculated using the squared chord distances based on the percentages of all
pollen taxa. To evaluate the analogue quality, we calculated the squared chord distances
between the selected fossil pollen spectra since the last glacial maximum (*n*=65,
elevation higher than 3000 m a.s.l.; Cao et al., 2013) and the combined modern pollen
dataset on the TP. The square chord distances were calculated using the *MAT* function
of the *rioja* package (Juggins, 2018) in R.

## 4 Data description

The pollen assemblages of the new surface-sediment samples ($n$=90) are dominated by herbaceous pollen from alpine meadow, steppe, and desert sites on the TP. In contrast, arboreal pollen dominates the samples collected from forest, consisting mainly of *Pinus*, *Picea*, *Alnus*, *Tsuga*, *Juniperus*, *Betula*, and *Quercus* (Fig. 2). Additionally, there are evident regional peculiarities in its distribution (Fig. 2–4). Sites with Cyperaceae abundances >60% from alpine meadows are more common than other sites, whereas steppe regions are marked by higher percentages of Poaceae and *Artemisia*, typically exceeding 30% and 50, respectively. The distribution center of Amaranthaceae (> 30%) is generally located in desert (Fig. 2–4; Table 2).

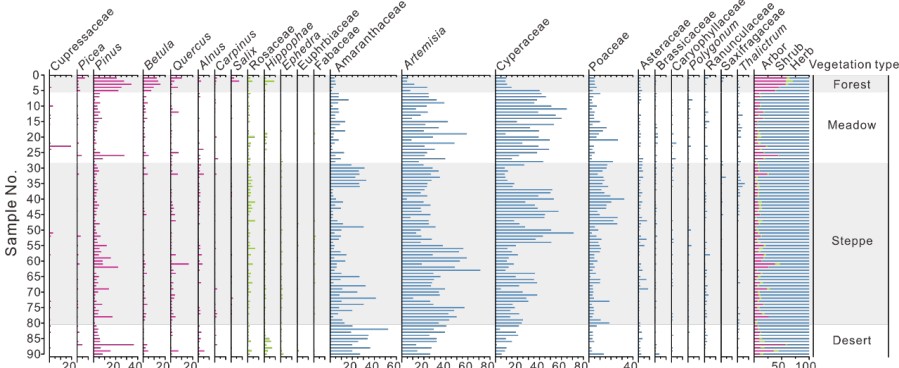

**Figure 2**. Percentage diagram of major pollen taxa for 90 lake surface-sediment samples on the Tibetan Plateau. Samples are arranged according to their vegetation type.

Group 1 (forest, $n$=5): The pollen assemblages of the sampled lakes are characterized by the lowest *Artemisia* and Amaranthaceae content, yet exhibits the highest arboreal pollen (AP) percentages among the four groups. Key arboreal taxa include *Pinus* (mean 26.0%, maximum 34.2%), *Betula* (mean 11.7%, maximum 15.6%), *Quercus* (mean 3.9%, maximum 9.3%), and *Picea* (mean 2.7%, maximum 7.0%, Fig. 2–4).

Group 2 (meadow, $n$=22): This group is typically characterized by the lowest AP and A/Cy (*Artemisia*/Cyperaceae) ratio but the highest Cyperaceae abundance (mean 39.8%, maximum 64.7%), with common taxa comprising *Artemesia* (mean 27.1%, maximum 58.9%), Amaranthaceae (mean 6.8%, maximum 16.4%), and Poaceae (mean 6.3%, maximum 26.1%, Fig. 2–4).

Group 3 (steppe, $n$=53): *Artemesia* (mean 28.9%, maximum 59.0%) is the most dominant component compared to meadow sites (Fig. 2–4). In addition, as a common taxon, Poaceae (mean 10.3%, maximum 31.4%), as well as the A/C (*Artemisia*/ Amaranthaceae (=Chenopodiaceae)) ratio (range 0.25–12.14, median 3.45) reach their highest values of the different vegetation types.



Group 4 (desert, *n*=10): These sites are characterized by the highest percentages of Amaranthaceae (mean 26.7%, maximum 52.4%), with higher *Artemisia* abundance (mean 27.4%, maximum 40.2%, Fig. 2–4), and the lowest Poaceae (mean 3.1%, maximum 6.6%), Cyperaceae (mean 11.4%, maximum 21.1%), and A/C ratio (range 0.55–2.08, median 0.83).

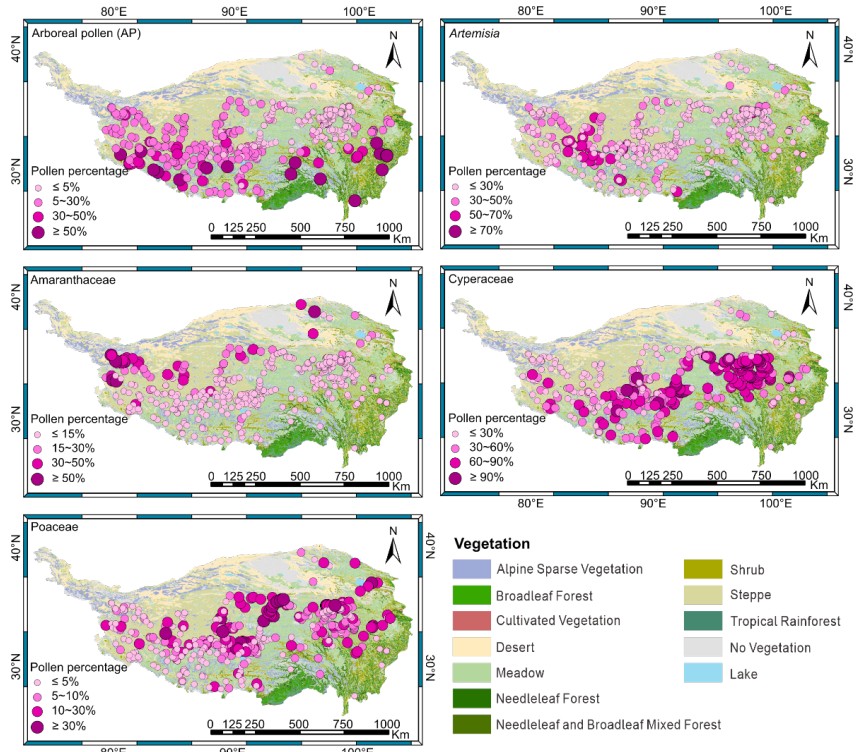

**Figure 3**. The spatial distribution maps of pollen percentages for total arboreal pollen (AP) and selected taxa (*Artemisia*, Amaranthaceae, Cyperaceae, Poaceae) in the dataset of lake surface-sediment samples (*n*=476) on the Tibetan Plateau.

Although AP pollen is detected at most meadow and steppe sites, and occasionally in desert regions, its abundance is markedly lower than in the forest sites (Table 1, Fig. 2–4). Since trees are absent in the alpine meadow, steppe, and desert communities on the TP (Wu, 1995; Wu and Xiao, 1995; Herzschuh et al., 2010), the low AP abundances likely represent wind-transported pollen transported from adjacent low-elevation regions. Despite this influence, the pollen assemblages effectively represent local vegetation composition, as the contribution of distant pollen is minimal overall (Fig. 2–4). Thus, the modern pollen distribution aligns closely with established vegetation types, corroborating findings from previous studies (Shen et al., 2006; Herzschuh et al., 2010;





Li et al., 2020). Pollen assemblages of the 476 pollen samples of the dataset from TP
are shown in Figure S1.

**Table 2.** Summary statistics of geophysical, climate variables, net primary production (NPP), and pollen percentages of the dataset on the Tibetan Plateau ($n$=476, Min: minimum; Med: median; Max: maximum, SD: standard deviation).

| Parameter | Min. | Med. | Max. | SD. | Pollen taxa | Min. | Med. | Max. | SD. |
|---|---|---|---|---|---|---|---|---|---|
| Longlitude | 79.36 | 91.76 | 102.55 | 91.88 | Ericaceae | 0.00 | 0.00 | 3.08 | 0.07 |
| Latitude | 27.62 | 33.13 | 39.36 | 32.93 | Euphorbiaceae | 0.00 | 0.00 | 30.94 | 0.23 |
| Elevation | 2668 | 4544 | 5433 | 4518 | Fabaceae | 0.00 | 0.20 | 5.53 | 0.40 |
| $Mt_{co}$ | -23.48 | -14.33 | -2.65 | -14.15 | Gentianaceae | 0.00 | 0.00 | 6.49 | 0.25 |
| $Mt_{wa}$ | 1.77 | 7.86 | 19.26 | 8.17 | *Hippophae* | 0.00 | 0.18 | 10.80 | 0.50 |
| $T_{ann}$ | -9.09 | -2.71 | 6.93 | -2.54 | Lamiaceae | 0.00 | 0.00 | 8.77 | 0.20 |
| $P_{ann}$ | 97 | 351 | 788 | 390 | *Picea* | 0.00 | 0.18 | 10.63 | 0.51 |
| NPP | 0.16 | 444.70 | 6617.36 | 660.54 | *Pinus* | 0.00 | 1.60 | 64.98 | 5.80 |
| Pollen taxa | Min. | Med. | Max. | SD. | Poaceae | 0.00 | 5.32 | 87.74 | 9.51 |
| *Abies* | 0.00 | 0.00 | 8.59 | 0.23 | Polemoniaceae | 0.00 | 0.00 | 15.21 | 0.09 |
| *Alnus* | 0.00 | 0.17 | 8.86 | 0.48 | *Polygonum* | 0.00 | 0.23 | 20.50 | 0.69 |
| *Artemisia* | 0.00 | 11.3 | 70.05 | 16.28 | *Quercus*_deciduous | 0.00 | 0.00 | 5.21 | 0.07 |
| Asteraceae | 0.00 | 1.70 | 33.56 | 16.28 | *Quercus*_evergreen | 0.00 | 0.00 | 27.81 | 1.17 |
| *Betula* | 0.00 | 0.36 | 30.59 | 1.49 | Ranunculaceae | 0.00 | 1.00 | 84.46 | 0.01 |
| Brassicaceae | 0.00 | 0.37 | 28.17 | 0.96 | *Rheum* | 0.00 | 0.00 | 3.73 | 2.53 |
| Caryophyllaceae | 0.00 | 0.30 | 10.79 | 0.59 | Rosaceae | 0.00 | 1.00 | 17.41 | 0.30 |
| Amaranthaceae | 0.00 | 2.08 | 64.89 | 6.54 | *Salix* | 0.00 | 0.00 | 7.16 | 1.76 |
| Crassulaceae | 0.00 | 0.00 | 2.49 | 0.06 | *Thalictrum* | 0.00 | 0.63 | 12.05 | 0.33 |
| Cupressaceae | 0.00 | 0.00 | 88.50 | 0.83 | Saxifragaceae | 0.00 | 0.00 | 4.69 | 1.10 |
| Cyperaceae | 0.00 | 38.37 | 96.68 | 43.01 | Thymelaeaceae | 0.00 | 0.00 | 8.33 | 0.12 |
| *Ephedra* | 0.00 | 0.15 | 7.45 | 0.36 | *Tsuga* | 0.00 | 0.00 | 6.47 | 0.03 |
| Ericaceae | 0.00 | 0.00 | 3.08 | 0.07 | *Urtica* | 0.00 | 0.00 | 3.87 | 0.23 |

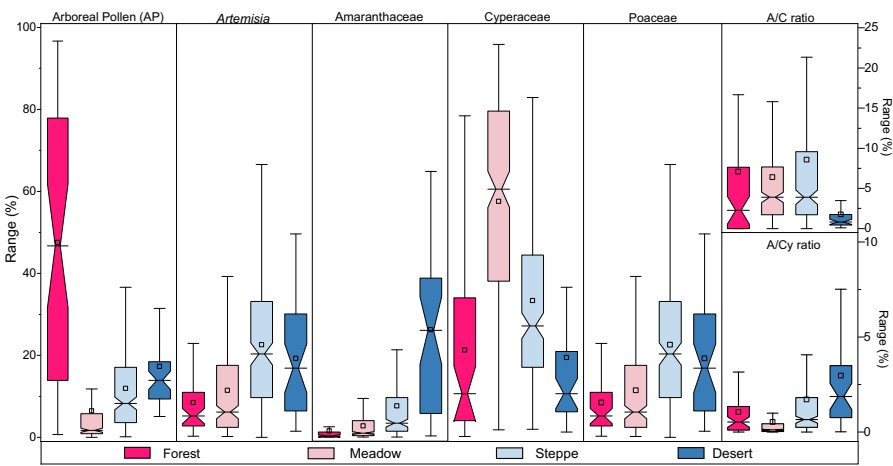

266





**Figure 4**. Box plots of the regional percentage distributions of arboreal pollen (AP) and four selected pollen types (*Artemisia*, Amaranthaceae, Cyperaceae, Poaceae), plus the ratios of A/C (*Artemisia*/Amaranthaceae (=Chenopodiaceae)) and A/Cy (*Artemisia*/Cyperaceae) from modern lake surface-sediment samples across the Tibetan Plateau.

The initial RDA indicated variance inflation factor (VIF) values exceeding 20 for the variables $T_{ann}$, $Mt_{co}$, and $Mt_{wa}$. However, after deleting $T_{ann}$, which had the highest VIF value, the remaining four variables ($P_{ann}$, $Mt_{co}$, $Mt_{wa}$, and NPP) had VIF values lower than 20. Thus, they are used in the final RDA to discern their influence on the modern pollen dataset.

**Table 3.** Summary statistics of redundancy analysis (RDA) of 476 sites, 35 pollen types, and four climatic variables ($P_{ann}$: mean annual precipitation, mm; $Mt_{co}$: mean temperature of the coldest month, °C; $Mt_{wa}$: mean temperature of the warmest month, °C; $T_{ann}$: annual mean temperature, °C) and NPP (net primary production) in the pollen dataset from the Tibetan Plateau. VIF: variance inflation factor.

| Climatic variables | VIF (without $T_{ann}$) | VIF (with $T_{ann}$) | $\lambda_1/\lambda_2$ | Climatic variables as sole predictor Explained variance (%) | Marginal contribution based on climatic variables Explained variance (%) | P-value |
|---|---|---|---|---|---|---|
| NPP | 1.94 | 2.19 | 0.21 | 7.29 | 0.67 | 0.006 |
| $P_{ann}$ | 3.10 | 3.43 | 0.44 | 13.13 | 3.92 | 0.001 |
| $Mt_{co}$ | 2.84 | 80.97 | 0.09 | 3.37 | 2.70 | 0.001 |
| $Mt_{wa}$ | 2.90 | 41.11 | 0.15 | 5.04 | 1.03 | 0.001 |
| $T_{ann}$ | — | 185.28 | — | — | — | — |

The RDA results highlight that, as a sole predictor, relative to $Mt_{co}$ and $Mt_{wa}$, $P_{ann}$ and NPP explain ubstantial portions of pollen assemblage variation (13.13% and 6.97%, respectively) in the dataset (Table 3). A biplot of the RDA shows that the vectors for both $P_{ann}$ and NPP form smaller angles with the positive direction of axis 1 (capturing 34.7% of total inertia in the dataset) compare to axis 2 (15.7%), suggesting moisture availability as the primary determinant along axis 1 (Fig. 5). RDA axis 1, which is highly correlated with NPP and $P_{ann}$, generally divides the pollen taxa into two groups. One group, comprising Cyperaceae, Ranunculaceae, and *Salix*, indicates wet climatic conditions (located along the positive direction of $P_{ann}$), while the other group, consisting of *Artemisia*, Amaranthaceae, Poaceae, *Ephedra*, and Saxifragaceae represents drought (located along the negative direction of $P_{ann}$; Fig. 5).

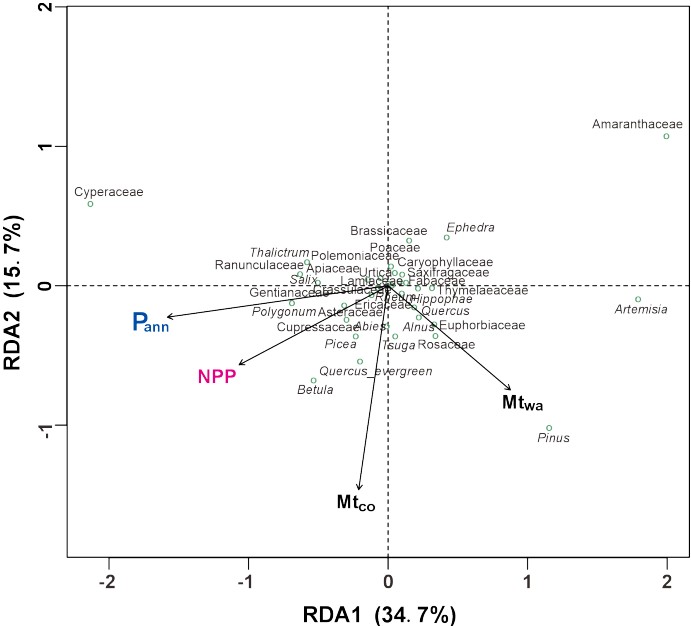

293

**Figure 5**. Redundancy analysis (RDA) biplot of the pollen dataset based on the first two axes showing the relationships between 35 pollen taxa (circles) and four variables (arrows) ($P_{ann}$: mean annual precipitation, mm; $Mt_{co}$: mean temperature of the coldest month, °C; $Mt_{wa}$: mean temperature of the warmest month, °C, and NPP: net primary production, Kg C m$^{-2}$).

## 5   Potential use of the lake surface-sediment pollen dataset

In the calibration-sets, $P_{ann}$ and NPP are selected as the target variables, as their identified importance in influencing pollen distribution, with NPP further providing insights into alpine vegetation conditions. The pollen-based modern $P_{ann}$ and NPP estimations using both WA-PLS and RF approaches match original measurements well, with a high coefficient of determination between observed and predicted variables ($R^2$) and low root mean square error of prediction (RMSEP) (Fig. 6). the RF model showed superior predictive performance compared to WA-PLS for both target variables.

Reconstructions covering $P_{ann}$ of ca. 300–600 mm and NPP lower than 1000 Kg C m$^{-2}$ should be reliable because their bias is low (Fig. 6). For $P_{ann}$, the proportion of residuals between -50 and 50 mm derived from RF (48.1%) is slightly higher than that of WA-PLS (45.6%). Similarly, for the range of -100 to 100 mm, RF (71.8%) outperforms WA-PLS (65.8%). For NPP, RF also shows a notably higher proportion of residuals between -500 and 500 Kg C m$^{-2}$ (84.5%) compared to WA-PLS (74.8%). This advantage persists for the narrower range of -300 to 300 kg C m$^{-2}$ (RF: 63.9% vs. WA-PLS: 50.4%). However, both models consistently overestimated Pann and NPP in arid areas with low productivity and underestimated these variables in humid, high-

productivity areas, highlighting the necessity of addressing the "edge-effect" (Fig. 6, 7).

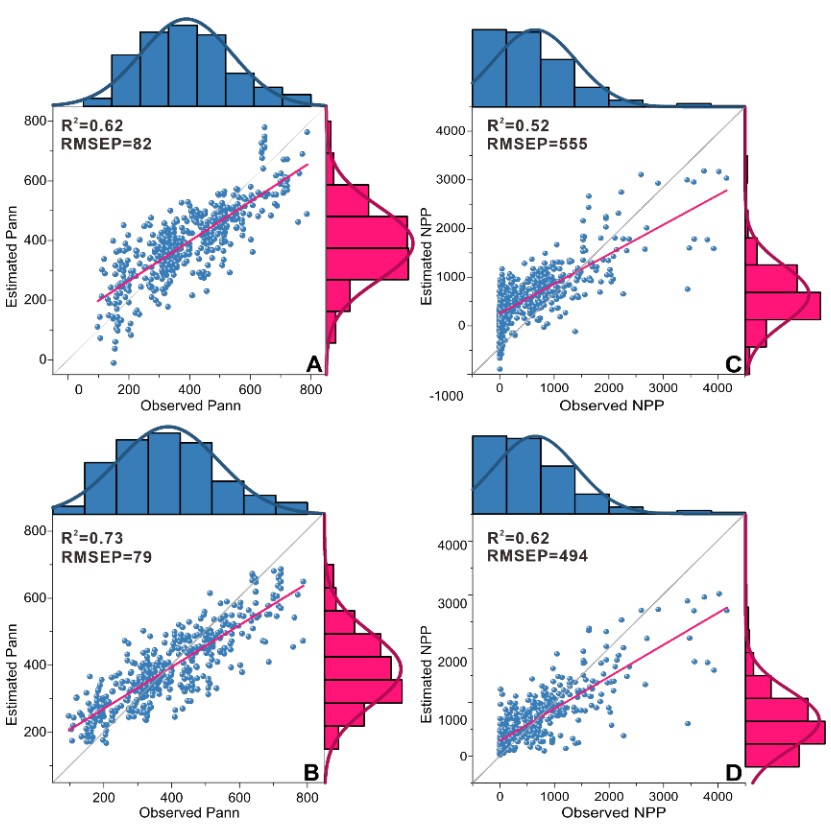


**Figure 6**. Scatter plots of observed mean annual precipitation ($P_{ann}$) vs. predicted $P_{ann}$, observed net
primary production (NPP) vs. predicted NPP using weighted-averaging partial least squares
regression (WA-PLS: top row) and random forest (RF: bottom row) based on the pollen data ($n$=476)
from lake surface-sediments on the Tibetan Plateau ($R^2$: coefficient of determination between
observed and predicted values; RMSEP: root mean square error of prediction produced by "leave-
one-out" cross-validation).

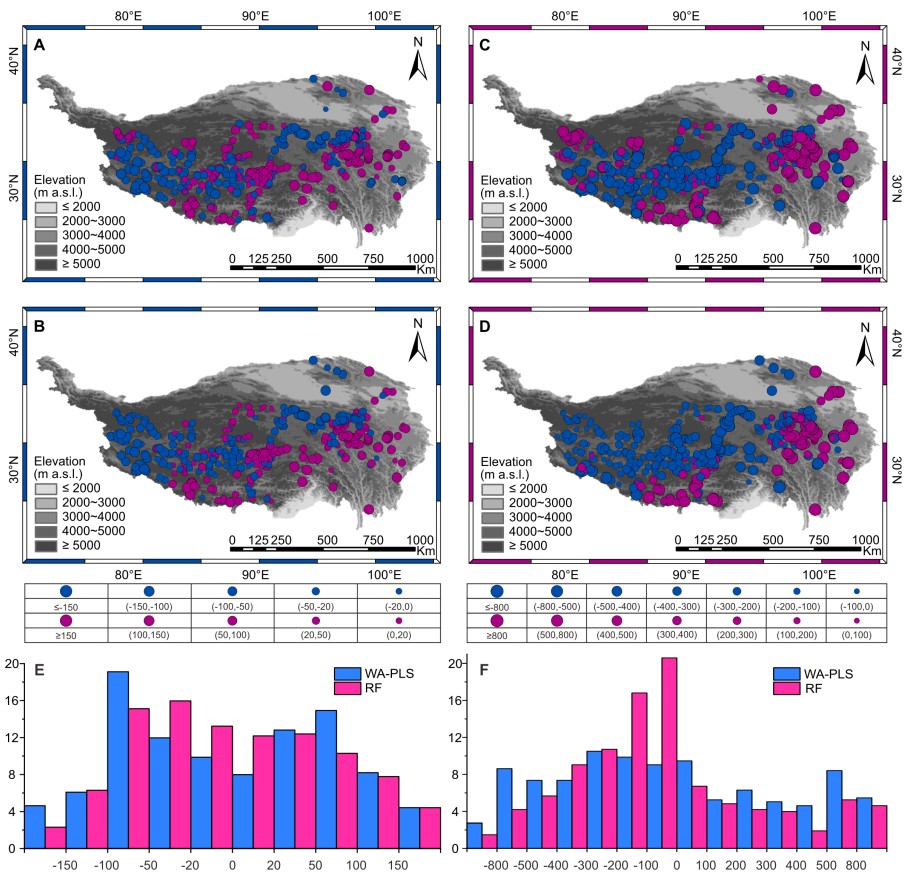

**Figure 7**. The residuals between observations and pollen-based reconstructions for the lake surface-sediment sites (*n*=476) on the Tibetan Plateau: (A) mean annual precipitation ($P_{ann}$) by weighted-averaging partial least squares regression (WA-PLS) and (B) random forest (RF), (C) net primary production (NPP) by WA-PLS and (D) RF. The two bar charts in the lower part of the figure show the proportions of modern pollen sites available within different ranges of residuals (observation minus reconstruction) for both $P_{ann}$ (E) and NPP (F).

Most of the poor analogue assemblages come from the TP margin and date back to >12 cal ka BP, which is possibly related to the higher abundance of arboreal pollen in this specific period and region (Fig. 8). While our combined modern pollen dataset from lake surface-sediments can provide good analogues for fossil pollen assemblages and enhance the performance of palaeoclimate reconstructions on the central TP, caution remains warranted for interpreting pollen assemblages from plateau margins and periods earlier than the Holocene (Fig. 8).


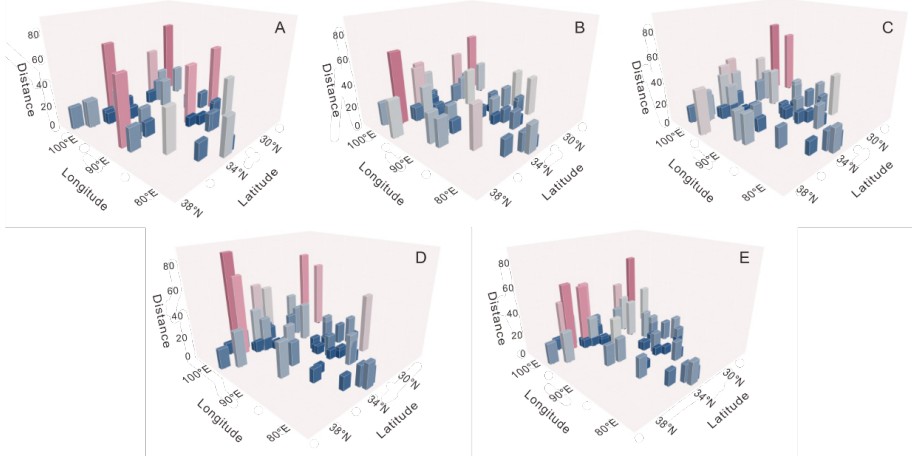

**Figure 8**. Spatial distribution of analogue quality for six key time slices on the Tibetan Plateau: (A) 15–12 cal ka BP; (B) 12–9 cal ka BP; (C) 9–6 cal ka BP; (D) 6–3 cal ka BP; (E) 3–0 cal ka BP.

## 6    Data availability

The modern pollen dataset from lake surface-sediment samples ($n$=90) comprising pollen percentages, site locations, net primary production, and climatic data for each lake is accessible from the National Tibetan Plateau / Third Pole Environment Data Center (TPDC; Tian, 2025; https://doi.org/10.11888/Paleoenv.tpdc.302470).

## 7    Summary

We established a comprehensive modern pollen dataset extracted from lake surface-sediments in forest, meadow, steppe, and desert vegetation types on the Tibetan Plateau by combining new modern pollen data with previous datasets. Numerical analyses reveal that mean annual precipitation ($P_{ann}$) is the most important climatic determinant influencing pollen distribution. Our dataset has good predictive power for past net primary production (NPP) and $P_{ann}$ reconstructions. The random forest algorithm is found to be a reliable approach for pollen-based reconstructions of past environments.

The pollen data from our sampled lakes help to fill the geographical gap left by previously published modern pollen datasets, thereby improving the spatial distribution of sampling sites covering the Tibetan Plateau. Our dataset is a key component for providing quantitative estimates of past vegetation or climate, and can also be integrated with other pollen datasets in the future to improve the reliability of past ecosystem and climate reconstructions on the Tibetan Plateau.

**Author contributions.** FT designed the pollen dataset, FT, WC, XC collected the samples, WC performed pollen extraction and identification. XC and FT compiled the



standardization for the dataset, performed numerical analyses, and organized the
manuscript. WC prepared the figures and tables. All authors discussed the results and
contributed to the final paper.

**Competing interests.** The corresponding author declares that none of the authors has
any competing interests.

**Disclaimer.** Publisher's note: Copernicus Publications remains neutral with regard to
jurisdictional claims in published maps and institutional affiliations.

**Acknowledgements.** The authors would like to express their gratitude to the
palynologists Ulrike Herzschuh (Alfred Wegener Institute Helmholtz Center for Polar
and Marine Research), Chunhai Li (Nanjing Institute of Geography and Limnology,
Chinese Academy of Sciences), Kai Li (College of Life Sciences, Zhejiang Normal
University), Qingfeng Ma (Institute of Tibetan Plateau Research, Chinese Academy of
Sciences) who contributed to the dataset. We thank Zhitong Chen (Institute of Tibetan
Plateau Research, Chinese Academy of Sciences), and students Meijiao Chen, Yunqing
Li, and Anjing Jian for their help with sample collections in the field work, and Cathy
Jenks with the help of language editing.

**Financial support.** This research was supported by the National Natural Science
Foundation of China (Grant No. 42471179, 42071107).

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
