# Peer review of "A lacustrine surface-sediment pollen dataset covering the Tibetan"

_Earth System Science Data, 2025_

## Author Response (AR1)

Dear Editor,

Herewith I resubmit our revised manuscript entitled "A lacustrine surface-sediment

pollen dataset covering the Tibetan Plateau and its potential in past vegetation and

climate reconstructions" (MS No.: essd-2025-242).

We would like to sincerely thank the two reviewers for their valuable comments on

our manuscript, which are helpful in improving our manuscript. We carefully

considered and responded to all comments and revised our manuscript accordingly.

We hope these changes will strengthen our manuscript. And thank you for

considering our manuscript and look forward to your response.

On behalf of the co-authors,

Yours faithfully,

Dr. Fang Tian,

College of Resource Environment and Tourism

Capital Normal University

Beijing, China

**105 West Third Ring Road North, Haidian District, Beijing, China 100048**

E-mail: tianfang@cnu.edu.cn

**Response to reviewer's comments**

**Referee #1**

Palynologists believe that vegetation and climate reconstructed from modern pollen of lake surface sediments is more reliable than from that of surface soils, but the number of modern pollen dataset from lake surface sediments is much less than that from surface soils. More modern pollen dataset from lake surface sediments is urgently needed. This study presents a lacustrine pollen dataset from many lake surface sediments covering the most part of the Tibetan Plateau. The modern pollen and climate relationships have been further analyzed. The dataset is well described and the statistical analyses are sound. There are however a couple of key problems should be addressed before publication.

**Specific comments:**

**1. Abstract**

(1) As the modern pollen data from 90 lakes are new, their detailed information should be provided, including the lake area and mean depth etc. The reviewer thinks that this information really matters the committing of vegetation and climate reconstructions. Meanwhile, the author provided 90 lake surface samples, whiles the analysis was based on 476 samples. This should be clarified in the abstract, and provide briefly the improvement in the main text.

Our response: We appreciate the reviewer's suggestion. In Table 1, we have added the water area (m²) of the 90 lakes, since it is a key factor influencing pollen source areas. Unfortunately, water depth data are not available because these samples were collected by different research teams. Nevertheless, we believe that water depth has only a limited impact on the representation of terrestrial pollen in lake surface sediments, and its absence does not affect the robustness of our analyses. In addition, we have clarified in the Abstract and Methods that the newly collected 90 lake-surface samples were used for descriptive purposes, while the integrated dataset of 476 samples (including both new and previously published data) was

employed for statistical analyses and for characterizing the spatial patterns of pollen distribution across the TP.

**Line 13–22:**

"We collected 90 new lake surface-sediment pollen samples from the Tibetan Plateau (TP), covering major vegetation types, including alpine forest, alpine meadow, alpine steppe, and alpine desert. By integrating these new data with previously published lacustrine pollen datasets, we established a comprehensive modern pollen dataset comprising 476 samples across the TP, covering the full range of climatic gradients across the TP, with net primary production (NPP) from 0.16 to 6617.36 Kg C  $m^{-2}$ , mean annual precipitation ( $P_{ann}$ ) from 97 to 788 mm, mean annual temperature ( $T_{ann}$ ) -9.09 to 6.93 °C, mean temperature of the coldest month ( $M_{tco}$ ) -23.48 to -2.65°C, and mean temperature of the warmest month ( $M_{twa}$ ) 1.77 to 19.26°C."

**Line 23-25:**

"Numerical analyses based on the comprehensive modern pollen dataset (n=476) revealed that  $P_{ann}$  is the primary climatic determinant for pollen distribution, while NPP is a valuable variable reflecting vegetation conditions."

**Line 145–149:**

"The final dataset comprises 476 pollen assemblages from lake surface-sediments on the TP (Fig. S1). The pollen assemblages of the 386 previously published samples have already been described and discussed in detail in their original publications. Therefore, in this study, we present only the pollen assemblages of the 90 newly collected samples." Line 173–175:

"For all statistical analyses (redundancy analysis: RDA, weighted averaging partial least squares regression: WA-PLS, and Random Forest: RF), we used the full integrated dataset of 476 samples."

**Line 176-178:**

"To visualize how the modern pollen assemblages respond to climatic variables, ordination techniques were employed based on the selected 35 pollen types from all 476 sites."

(2) Line 14, covering major vegetation types. Please list the major vegetation types from the whole dataset, including the later added previous modern lacustrine pollen dataset. This is because later the authors only presented the climate range but no vegetation information. The final utilization of the dataset is to reconstruct both the climate and vegetation, but the vegetation-pollen relationship has not been investigated in this study.

Our response: We thank the reviewer for this helpful suggestion. In the revised version, we have explicitly listed the major vegetation types represented in the whole dataset, including those from the later-added lacustrine pollen dataset. We also added RDA results (Fig. 5b and related text) based on pollen sites, and clarified the relationship between pollen and NPP (an indicator of vegetation condition) in the introduction. These changes should make the vegetation information more complete and better balance the climate and vegetation perspectives.

**Line 13–15:**

"We collected 90 new lake surface-sediment pollen samples from the Tibetan Plateau (TP), covering major vegetation types, including alpine forest, alpine meadow, alpine steppe, and alpine desert."

**Line 72–78:**

"However, the pollen concentration and percentages from lake sediments have been confirmed to positively correlated with vegetation coverage, which reflects total plant biomass (Liu et al., 2023). Since net primary production (NPP) represents the carbon fixed and accumulated as biomass by plants (Fang et al., 2001; Nemani et al., 2003; Gonsamo et al., 2013; Ni, 2013; Walker et al., 2015; Ji et al., 2020), pollen can serve as an indirect proxy for NPP, allowing us to infer spatial and temporal patterns of vegetation conditions on the TP."

**Line 310-313:**

"Furthermore, samples collected from alpine desert, steppe, meadow, and forest are located along the gradients of NPP and  $P_{ann}$  (Fig. 5B), indicating that they can effectively distinguish different vegetation types as well as pollen assemblages."

(3) Line 15-17, the total number of pollen sites (476) should be provided here.

Our response: Agreed and done.

month (Mtwa) 1.77 to 19.26°C."

Line 15-22:

"By integrating these new data with previously published lacustrine pollen datasets, we established a comprehensive modern pollen dataset comprising 476 samples across the TP, covering the full range of climatic gradients across the TP, with net primary production (NPP) from 0.16 to 6617.36 Kg C  $m^{-2}$ , mean annual precipitation ( $P_{ann}$ ) from 97 to 788 mm, mean annual temperature ( $T_{ann}$ ) -9.09 to 6.93 °C, mean temperature of the coldest month ( $Mt_{co}$ ) -23.48 to -2.65°C, and mean temperature of the warmest

2. Study area

(1) There are totally 476 modern lake pollen samples, covering most of the TP, but there are still some geographical gaps in northern, western (the driest region) and southeastern TP (the moistest region). Do the authors have any suggestions to get more samples from these regions?

Our response: Due to the geographical difficulties, we did not collect samples from the northwestern Tibetan Plateau. In the revised version, we have added a note that future work should focus on supplementing pollen samples from northern, western, and southeastern regions to improve spatial coverage. However, the 90 samples collected from major vegetation types and climate gradients are highly valuable for detecting modern pollen–vegetation, pollen–climate relationships, and for providing a robust database to quantitively reconstruct the past vegetation and climate changes.

Line 11–13:

"A dataset of pollen extracted from the surface-sediments of lakes with broad spatial coverage is essential for pollen-based reconstructions of past vegetation and climate." Line 117–122:

"To achieve a broadly representative coverage of lakes across different vegetation zones on the TP, we collected one surface-sediment sample (top 2 cm) from the centre of each

lake, for a total of 90 lakes across different vegetation types on the TP: forest (n=5), meadow (n=22), steppe (n=53), and desert (n=10) between 2021 and 2023 (Fig. 1, Table 1). Collecting from the lake centre is intended to provide a representative pollen assemblage that integrates inputs from the surrounding catchment."

**Line 375–377:**

"Moreover, the current spatial coverage of lakes across the TP is still not fully even, highlighting the need for additional sampling to achieve a more representative dataset in future work."

(2) Another question: how many pollen samples from one lake? Only one or several? If only one sample, what is the difference from big lake and small lake? In the method section, the authors sad that there is only one sample for the 90 newly collected pollen samples. Is the one sample representative for some of the big lakes?

Our response: While this approach is commonly used in modern pollen dataset studies, we acknowledge that a single sample may not fully capture within-lake spatial heterogeneity, particularly in large lakes. The potential influence of lake size on pollen assemblages is difficult to disentangle from the effects of surrounding vegetation composition and community structure. We have added a sentence in the Methods section to clarify our sampling strategy and a note in the Discussion to acknowledge this limitation and potential source of uncertainty. In this study, only one pollen sample from one lake was used, which was collected from the lake centre, normally to obtain a representative assemblage that integrates pollen input over a broader area.

**Line 117-122:**

"To achieve a broadly representative coverage of lakes across different vegetation zones on the TP, we collected one surface-sediment sample (top 2 cm) from the centre of each lake, for a total of 90 lakes across different vegetation types on the TP: forest (n=5), meadow (n=22), steppe (n=53), and desert (n=10) between 2021 and 2023 (Fig. 1, Table 1). Collecting from the lake centre is intended to provide a representative pollen assemblage that integrates inputs from the surrounding catchment."

(3) Fig 1, where is the NPP data come from? Reference related to the NPP data should be cited. Can the vegetation division of the TP be added in the figure or in a new figure? Our response: Agreed and done. We added the NPP data source and a new figure in Fig.1 to present the vegetation division of TP.

**Line 95–98:**

"Figure 1. Spatial distribution of 476 modern pollen samples collected from lake surface-sediments on the Tibetan Plateau (red filled circles: 90 sampled lakes; orange filled circles: 386 previous samples; Herzschuh et al., 2010; Li and Li, 2015; Cao et al., 2021; Ma et al., 2024; Wu et al., 2024) based on (A) vegetation types and (B) net primary production (NPP, Zhao and Running, 2010)."

**3. Methods**

(1) Line 114, To ensure the even distribution of the sampled lakes, we collected ... However, Figure 1 shows that the sampled lakes are not evenly distributed as some points are crowned with each other.

Our response: Agreed. We agree that the spatial distribution of the 476 sampled lakes is not perfectly even, as some lakes are clustered in certain regions. We have revised the text in the Methods section to more accurately reflect the distribution and clarify that the sampling aimed to broadly represent major vegetation and climate gradients across the TP.)

**Line 117–122:**

"To achieve a broadly representative coverage of lakes across different vegetation zones on the TP, we collected one surface-sediment sample (top 2 cm) from the centre of each lake, for a total of 90 lakes across different vegetation types on the TP: forest (n=5), meadow (n=22), steppe (n=53), and desert (n=10) between 2021 and 2023 (Fig. 1, Table 1). Collecting from the lake centre is intended to provide a representative pollen assemblage that integrates inputs from the surrounding catchment."

(2) Line 141-142, The pollen data are standardized following the procedures outlined in Cao et al. (2013), including harmonization of taxonomy – generally to the family or genus level. Such harmonization of pollen taxonomy might loss some information, can the original pollen taxa be published in the dataset? For the statistical analyses, however, such harmonization is fine.

Our response: Agreed and done. We note that the original surface-sediment pollen data from previous studies have already been published by the respective authors, and we have cited them accordingly. For statistical analyses, all pollen data were harmonized to family or genus level following Cao et al. (2013). To minimize information loss, however, we added the original taxonomic names in the revised dataset for the 90 newly collected samples.

(3) Line 144-145, a maximum ≥3% were retained for statistical analyses (n=35). What does this mean? On the other hand, the authors mentioned all pollen taxa after. Please clarify is these 35 taxa was used in RDA, WAPLS and RF.

Line 150-153, line 163-164, line 189-190: the same question as mentioned above. The analysis was based on all samples, all pollen taxa, or selected 35 taxa? Please verify.

Our response: Agreed and done. In this study, we clarified that the selected 35 pollen types across all 476 pollen samples were applied in RDA, WAPLS and RF. Line 152–155:

"Only pollen taxa with an abundance of at least 0.5% in at least three samples and a maximum  $\geq$ 3% (n=35) were retained for the following statistical analyses (RDA, WA-PLS, and RF)."

**Line 160-163:**

"For all the 476 lakes, the following parameters were extracted:  $P_{ann}$ : mean annual precipitation, mm;  $T_{ann}$ : mean annual temperature, °C;  $Mt_{co}$ : mean temperature of the coldest month, °C;  $Mt_{wa}$ : mean temperature of the warmest month, °C (He et al., 2020)." Line 176–178:

"To visualize how the modern pollen assemblages respond to climatic variables, ordination techniques were employed based on the selected 35 pollen types from all 476 sites."

**Line 213-215:**

"The RF algorithm was run based on square-root transformed pollen percentages of the selected 35 taxa, using the randomForest function in the randomForest package version 4.6–14 (Liaw, 2018) in R."

(4) Line 170, along climate gradients. The NPP was also used as one parameter, but it is not a climatic variable.

Our response: Agreed and done.

**Line 182-184:**

"We employed RDA to assess how major pollen taxa and sampling sites are distributed along vegetation and climate gradients."

(5) Line 209-210, based on the percentages of all pollen taxa. All taxa or 35 selected taxa??

**Our response**: In the evaluation of analogue quality, we used all pollen taxa to match the fossil pollen data and modern pollen data.

**4. Data analysis**

(1) The pollen-climate relationships and the reconstruction of climate based on modern pollen data have been extensively studied using very complex methods in this manuscript, but no any study about the pollen-vegetation relationships and the reconstruction of vegetation based on pollen data. The reviewer suggests that reducing the content of climate study but add some study on the vegetation reconstruction.

Our response: We thank the reviewer for this valuable suggestion. In fact, our original manuscript already included vegetation-related analyses. We used NPP as an indicator of vegetation conditions and examined the modern pollen—NPP relationships. Furthermore, the RDA results also clarify the relationships between

modern pollen assemblages and vegetation distribution. These analyses complement the pollen-climate reconstructions and ensure that both vegetation and climate aspects are considered in this study.

Line 310-313:

"Furthermore, samples collected from alpine desert, steppe, meadow, and forest are located along the gradients of NPP and  $P_{ann}$  (Fig. 5B), indicating that they can effectively distinguish different vegetation types as well as pollen assemblages."

(2) Data description

Line 250, should "selected herbaceous taxa" be more precise?

Our response: Agreed and done.

Line 254–256:

"Figure 3. The spatial distribution maps of pollen percentages for total arboreal pollen (AP) and selected herbaceous taxa (Artemisia, Amaranthaceae, Cyperaceae, Poaceae) in the dataset of lake surface-sediment samples (n=476) on the Tibetan Plateau." Line 279–282:

"Figure 4. Box plots of the regional percentage distributions of arboreal pollen (AP) and four selected herbaceous pollen types (Artemisia, Amaranthaceae, Cyperaceae, Poaceae), plus the ratios of A/C (Artemisia/Amaranthaceae (synonym: Chenopodiaceae)) and A/Cy (Artemisia/Cyperaceae) from modern lake surface-sediment samples across the Tibetan Plateau."

(3) Only the newly collected pollen sites have been analyzed in their pollen features. Why not analyze the whole dataset with 476 pollen samples?

Our response: The descriptive analysis focuses on the 90 newly collected samples, which are broadly representative of the spatial patterns and pollen assemblage characteristics across the Tibetan Plateau, as they cover all major vegetation types and exhibit similar dominant taxa to the full dataset. Additionally, the previously published 386 samples have already been described in detail in the literature (Herzschuh et al., 2010; Li and Li, 2015; Cao et al., 2021...), so re-describing them here would be redundant. However, all statistical analyses (RDA, WA-PLS, and

Random Forest) were conducted using the full dataset of 476 samples. This approach highlights the new sites while maintaining the comprehensiveness of the analyses. In addition, we have clarified in Methods.

**Line 147-149:**

"The pollen assemblages of the 386 previously published samples have already been described and discussed in detail in their original publications. Therefore, in this study, we present only the pollen assemblages of the 90 newly collected samples."

**Line 173–175:**

"For all statistical analyses (redundancy analysis: RDA, weighted averaging partial least squares regression: WA-PLS, and Random Forest: RF), we used the full integrated dataset of 476 samples."

(4) How can use pollen to indicate NPP? In my thought, NPP is the indicator of vegetation rather than species or a group of species?

Our response: Pollen is a reliable proxy to indicate plant community composition. Previous studies confirmed that both of pollen percentage and concentration correlated with vegetation coverage. Vegetation coverage is closely linked to community productivity, pollen data can therefore serve as an indirect indicator of net primary production (NPP).

**Line 72–78:**

"However, the pollen concentration and percentages from lake sediments have been confirmed to positively correlate with vegetation coverage, which reflects total plant biomass (Liu et al., 2023). Since net primary production (NPP) represents the carbon fixed and accumulated as biomass by plants (Fang et al., 2001; Nemani et al., 2003; Gonsamo et al., 2013; Ni, 2013; Walker et al., 2015; Ji et al., 2020), pollen can serve as an indirect proxy for NPP, allowing us to infer spatial and temporal patterns of vegetation conditions on the TP."

**(5) Technical corrections**

Line 56, delete.

Line 139, where is the Fig A1? Should be Fig S1?

Line 134, Tibetan Plateau – TP

Line 253, than that in forest sites

Line 313, Pann, lower case

Our response: Agreed and done.

Line 55–59:

"Fortunately, the widespread distribution of lakes across the plateau offers an opportunity to expand and refine pollen-based calibration datasets using lake surface sediments, but the distribution of available pollen sites of lake surface-sediment remains uneven and incomplete due to logistical constraints (Cao et al., 2021; Qin,

2021; Ma et al., 2024)."

Line 140–143:

"We compiled a dataset of modern pollen assemblages from lake surface sediments across the TP, incorporating 375 lakes situated in the eastern (Herzschuh et al., 2010; Cao et al., 2021), central, and western TP (Ma et al., 2024; Wu et al., 2024), obtained from accessible databases or from authors directly."

Line 145–146:

"The final dataset comprises 476 pollen assemblages from lake surface-sediments on the TP (Fig. S1)."

Line 267–269:

"Although AP pollen is detected at most meadow and steppe sites, and occasionally in desert regions, its abundance is markedly lower than that in the forest sites (Table 1, Figs. 2–4)."

Line 336-339:

"However, both models consistently overestimated NPP and  $P_{ann}$  in arid areas with low productivity and underestimated these variables in humid, high-productivity areas, highlighting the necessity of addressing the "edge-effect" (Figs. 6, 7)."

**Referee #2**

This work collected 90 lake surface-sediment samples and integrated 375 previous data from the Tibetan Plateau (TP) covering major vegetation types. At the same time, this word detected the quantitative relationship between pollen and environmental factors, such as NPP and precipitation, which is valuable for Tibetan Plateau studies.

I still have some minor revision comments:

1. I think the authors should give more information about the 90 sites. For example, whether authors did vegetation investigation during sampling? Except the vegetation type, whether authors could give more information about the vegetation around the sampling sites. I think these information is important for establishing the relationships between vegetation and pollen.

Our response: We agree that detailed vegetation data are valuable and important for investigating the relationships between modern pollen and vegetation and the reconstructions of past vegetation changes, but for lake sediments the pollen source areas usually cover the entire catchment and exceed the range of field surveys, making vegetation investigation at 90 sites unfeasible. Moreover, the lakes are commonly surrounded by diverse plant communities, and adequate vegetation investigation would require extensive sampling of multiple plots around each lake, which is not feasible for 90 lakes. For this reason, detailed vegetation surveys are not available. Instead, we provided lake water area information, and the integrated dataset of 476 sites still ensures robust pollenvegetation relationships across the TP.

2. I think the table 2 is not very important, which could be put in the supporting information or revised to a figure, because the min and max value is not very important for readers, who cares more about the range and value distribution frequency.

Our response: Agreed and done. Table 2 was put in the supporting information.

3. The figure 8 is a little hard to understand. I suggest to put these data in map, which

could be more visualized.

Our response: Agreed and done. We revised Fig. 8.

---

## Author Response (AR2)

**Response to Reviewer – General Revision Summary**

In this revision, we have made the following modifications to improve clarity, consistency, and accuracy throughout the manuscript:

- 1. We have standardized the use of abbreviations by spelling out the full terms only at their first appearance in the abstract and main text, as well as in all figure and table captions. And then reviewed the capitalization of proper nouns and technical terms for consistency (e.g., Weighted-Averaging Partial Least Squares (WA-PLS) and Random Forest algorithm (RF))
- 2. The use of hyphens, en dashes, and minus including the replacement of em dash have been standardized throughout the text. Several grammatical issues (e.g., subject–verb agreement such as was/were) have also been revised.
- 3. We have clarified the usage of Amaranthaceae and Chenopodiaceae in this study to ensure consistency with current taxonomic conventions.
- 4. In Section 4: Data Description, the format for presenting the mean and maximum values of pollen percentages has been revised for improved clarity and conciseness.
- 5. Finally, several references have been corrected in terms of formatting and citation style.
- 6. For figure and table captions,
  - a) In Figure 1, we corrected the number of previously published samples indicated in the legend and the units of NPP value.
  - b) Table 1 has been revised to correct the names of several lakes.
  - c) In Figure 3, according to the revision of mentioned above (3.), we adjusted the 'Chenopodiaceae' to 'Amaranthaceae'.
  - d) In Figure 4, we revised the unit labels for the A/C and A/Cy ratios, removing the percentage symbol (%) to reflect the correct measurement unit.
  - e) In Figure 8, we adjusted the arrangement of the five sub-graphs and refined the font settings and other visual elements.

We hope these revisions address the concerns and improve the overall quality of the manuscript.